# GOLPH3 Participates in Mitochondrial Fission and Is Necessary to Sustain Bioenergetic Function in MDA-MB-231 Breast Cancer Cells

**DOI:** 10.3390/cells13040316

**Published:** 2024-02-08

**Authors:** Catalina M. Polanco, Viviana A. Cavieres, Abigail J. Galarza, Claudia Jara, Angie K. Torres, Jorge Cancino, Manuel Varas-Godoy, Patricia V. Burgos, Cheril Tapia-Rojas, Gonzalo A. Mardones

**Affiliations:** 1Centro de Biología Celular y Biomedicina (CEBICEM), Facultad de Medicina y Ciencia, Universidad San Sebastián, Providencia, Santiago 7510156, Chile; cmelendezp@correo.uss.cl (C.M.P.); viviana.cavieres@uss.cl (V.A.C.); cjcjarao@gmail.com (C.J.); aktorres@uc.cl (A.K.T.); jorge.cancino@uss.cl (J.C.); manuel.varas@uss.cl (M.V.-G.); patricia.burgos@uss.cl (P.V.B.); 2Departamento de Ciencias Biológicas y Químicas, Facultad de Medicina y Ciencia, Universidad San Sebastián, Campus Los Leones, Providencia, Santiago 7510156, Chile; 3Escuela de Medicina, Facultad de Medicina y Ciencia, Universidad San Sebastián, Valdivia 5110693, Chile; abigalarza98@gmail.com; 4Centro Científico y Tecnológico de Excelencia Ciencia & Vida, Fundación Ciencia & Vida, Huechuraba, Santiago 8580702, Chile; 5Centro de Excelencia en Biomedicina de Magallanes (CEBIMA), Universidad de Magallanes, Punta Arenas 6210427, Chile

**Keywords:** GOLPH3, PI(4)P, DRP1, mitochondrial bioenergetics, mitochondrial fission and fusion, mitochondrial fragmentation, Golgi apparatus, Golgi–mitochondria communication

## Abstract

In this study, we investigated the inter-organelle communication between the Golgi apparatus (GA) and mitochondria. Previous observations suggest that GA-derived vesicles containing phosphatidylinositol 4-phosphate (PI(4)P) play a role in mitochondrial fission, colocalizing with DRP1, a key protein in this process. However, the functions of these vesicles and potentially associated proteins remain unknown. GOLPH3, a PI(4)P-interacting GA protein, is elevated in various types of solid tumors, including breast cancer, yet its precise role is unclear. Interestingly, GOLPH3 levels influence mitochondrial mass by affecting cardiolipin synthesis, an exclusive mitochondrial lipid. However, the mechanism by which GOLPH3 influences mitochondria is not fully understood. Our live-cell imaging analysis showed GFP-GOLPH3 associating with PI(4)P vesicles colocalizing with YFP-DRP1 at mitochondrial fission sites. We tested the functional significance of these observations with GOLPH3 knockout in MDA-MB-231 cells of breast cancer, resulting in a fragmented mitochondrial network and reduced bioenergetic function, including decreased mitochondrial ATP production, mitochondrial membrane potential, and oxygen consumption. Our findings suggest a potential negative regulatory role for GOLPH3 in mitochondrial fission, impacting mitochondrial function and providing insights into GA–mitochondria communication.

## 1. Introduction

The functions of eukaryotic cells depend on the concerted activities of their organelles. Numerous instances of inter-organelle communication have been identified [1]. While the functional impact of several types of inter-organelle communication is fairly understood, communication between the Golgi apparatus (GA) and other organelles has been less characterized [2]. The most extensively studied inter-organelle communication involving the GA is with the endoplasmic reticulum (ER), which, for instance, ensures the synthesis and homeostasis of various biomolecules [2]. One of these processes involves the synthesis of ceramides in the ER, which are a family of lipid molecules that are transferred to the GA for subsequent conversion into complex sphingolipids [3]. This transfer can occur through vesicular transport or direct contact between these organelles [4]. Regardless of the inter-organelle communication mechanism, proteins serve as primary mediators of functional interactions between different organelles, and an increasing number of proteins participate in these processes, categorized by their functions as structural, recruitment, regulatory, and anchoring proteins, among others [5].

One of the least understood forms of inter-organelle communication is between the GA and mitochondria. To date, the functional significance of this interaction remains to be established. Yet, a connection between the GA and mitochondria was observed in pancreatic acinar cells almost two decades ago [6]. More recently, however, a potential functional interaction between the GA and mitochondria has been suggested during mitochondrial fission [7]. This interaction involves the transfer of vesicles containing phosphatidylinositol 4-phosphate (PI(4)P) from the *trans*-most cisternae of the GA to mitochondria fission sites [7]. Although their function is not fully understood, these vesicles arrive at these sites after the recruitment of Dynamin-related protein 1 (DRP1), a protein necessary for mitochondrial fission, suggesting the potential involvement of these vesicles in the final stages of this process [7]. These vesicles also appear to synergistically contribute to DRP1 activity through actin polymerization [8]. Nevertheless, the functional significance of these findings remains unclear, and the specific molecular players that mediate this inter-organelle interaction are unknown.

One protein candidate that could mediate communication between the GA and mitochondria is Golgi phosphoprotein 3 (GOLPH3). GOLPH3 is a peripheral protein of the GA with a large cytosolic pool [9,10,11]. It localizes to the GA as a result of its interaction with PI(4)P [12,13], a lipid enriched in the *trans*-most cisternae of the GA [14]. Importantly, GOLPH3 is found overexpressed in various types of solid tumors, including colorectal adenocarcinoma, prostate cancer, and breast cancer [15,16,17,18]. Moreover, in several types of cancer, GOLPH3 expression levels correlate with poor prognosis [19]. In contrast, GOLPH3 knockdown reduces tumorigenicity in various types of cancer cell lines [19]. However, it is not clear how the increased levels of GOLPH3 trigger transformation. Several functions have been proposed for GOLPH3, which are as diverse as maintaining GA structure [13,20], sustaining the localization of specific glycosyltransferases within the GA [21,22,23], cell survival after DNA damage [24], and regulation of focal adhesion dynamics [25], among other functions. To date, however, a precise picture of the molecular mechanisms that account for the diversity of GOLPH3 functions is missing. Intriguingly, GOLPH3 has also been associated with mitochondria. Early studies suggested that in HeLa cells, the levels of GOLPH3 affect mitochondrial mass [26]. Later, experimental overexpression of GOLPH3 in the metastatic human adenocarcinoma breast cancer cell line MDA-MB-231 was used to promote mitochondrial biogenesis to alter mitochondrial metabolism in cancer cells [27]. Thus, given the recent discovery that PI(4)P-containing GA vesicles associate with mitochondrial fission sites and that GOLPH3 is a PI(4)P-binding protein, we decided to investigate whether GOLPH3 is found in these vesicles and whether its levels affect mitochondrial fission and bioenergetic capacity in MDA-MB-231 cells. Our results suggest that GOLPH3 could be a novel negative regulator of mitochondrial fission, playing unexpected roles in maintaining mitochondrial dynamics and bioenergetic function.

## 2. Materials and Methods

### 2.1. DNA Constructs and Cell Reagents

The generation of the pEGFP mammalian expression construct encoding human wild-type GOLPH3 fused at the C-terminal of the green fluorescent protein (pGFP-GOLPH3-WT) was described elsewhere [10]. The mammalian expression construct encoding the PI(4)P reporter PH-FAPP1 (mCherry-PH-FAPP1) was a generous gift from Antonella de Matteis (Telethon Institute of Genetics and Medicine, Pozzuoli, Italy). The pEYFP-C1-DRP1 mammalian expression construct encoding human full-length DRP1 (YFP-DRP1) (Addgene, Watertown, MA, USA; cat # 45160) was a generous gift from Verónica Eisner (Pontificia Universidad Católica de Chile, Santiago, Chile). The mitochondrial fluorescent dyes MitoTracker^TM^ Deep Red FM, MitoTracker^TM^ Red CMXRos, and tetramethylrhodamine ethyl ester perchlorate (TMRE) were obtained from Thermo Fisher Scientific (Waltham, MA, USA; cat # M22426, cat # M7512, and cat # T669, respectively). The fluorescent nuclear stain 4′,6-diamidino-2-phenylindole (DAPI) was from Thermo Fisher Scientific (cat # D1306).

### 2.2. Cell Culture and Transfection

HeLa cells and the triple-negative breast cancer cell line MDA-MB-231 were obtained from the American Type Culture Collection (Manassas, VA, USA). HeLa cells were maintained in Dulbecco’s Modified Eagle’s Medium (DMEM) with high glucose (HyClone, Logan, UT, USA; cat # SH30343.02), and MDA-MB-231 cells were maintained in DMEM/F-12 (Thermo Fisher Scientific, cat # 12400-024). The media were supplemented with 10% (*v*/*v*) of heat-inactivated (56 °C for 30 min) fetal bovine serum (Biowest, Nuaillé, France; cat # S1810-500-B), 100 U/mL penicillin and 100 μg/mL streptomycin (Thermo Fisher Scientific; cat # 15140122). Cells were cultured in a humidified incubator with 5% (*v*/*v*) CO_2_ at 37 °C. Transient transfections were carried out using Lipofectamine^TM^ 2000 reagent (Thermo Fisher Scientific, cat # 11668-019) according to the manufacturer’s instructions, and cells were analyzed 16–24 h after transfection.

### 2.3. Time-Lapse Microscopy and Image Processing

Cells grown for 24 h on 35 mm glass-bottom culture dishes (MatTek, Ashland, MA, USA) were transiently co-transfected to express either GFP-GOLPH3 and mCherry-PH-FAPP1 or GFP-GOLPH3 and YFP-DRP1. After 16 h, cells were incubated with MitoTracker^TM^ Deep Red FM dye for 20 min at 37 °C. The culture medium was replaced with phenol red-free buffered DMEM medium (Thermo Fisher Scientific, cat # 21063-029) supplemented with 10% (*v*/*v*) of heat-inactivated (56 °C for 30 min) fetal bovine serum, 100 U/mL penicillin and 100 μg/mL streptomycin. Culture dishes with cells were held at 37 °C on a temperature-controlled microscope stage (UNO-temp controller, OKOLAB chamber; Pozzuoli, Italy) and 8-bit images (250–500 frames) were acquired every 0.8–1.0 s with a TCS SP8 laser scanning confocal microscope equipped with a PlanApo 63× oil immersion objective (NA 1.4); 405 nm, 488 nm, 561 nm and 643 nm lasers; an HyD hybrid detector system; and using LASX microscopy software (version 3.5.2.18963; Leica, Wetzlar, Germany). Selected images were processed for deconvolution using Huygens Essential software (version 23.10; Scientific Volume Imaging; Hilversum, The Netherlands). Processed images were visually inspected with ImageJ software (version 1.53f51v), which was also used to prepare movies and select single frames to prepare figures. We obtained at least 100 sets of time-lapse microscopy images per each condition.

### 2.4. Generation of CRISPR-Cas9-Mediated GOLPH3 Knockout Cells

Four guide RNA (gRNA) molecules (#1 GCTCCTTGTCGGGCGGCGTTGCGG; #2 CTCCTTGTCGGCGGCGTTGCGGG; #3 CTTGTCGGCGCGTTGCGGGAGG; #4 CGACGACAAGGGCGACTCCAAGG) targeting exon 1 of the human gene encoding GOLPH3 and the pBABE-Cas9 vector, providing resistance to puromycin, were a generous gift from Alejandro Rojas (Universidad Austral de Chile, Valdivia, Chile). To generate GOLPH3 knockout (KO) MDA-MB-231 cells, cells were seeded at low confluency in 6-well plates. After 48 h, cells were co-transfected with either of the four gRNA molecules in conjunction with the pBABE-Cas9 vector using Lipofectamine^TM^ 2000 reagent (Thermo Fisher Scientific, cat # 11668-019) according to the manufacturer’s instructions. After 72 h, transfected cells were selected by adding puromycin to a final concentration of 3 μg/mL. After 30 h, cells were transferred to 100 mm plates to select clones, which were subsequently subjected to limiting dilution in 96-well plates. KO testing was performed by immunofluorescence and immunoblot analysis with polyclonal rabbit anti-GOLPH3 antibody and homemade mouse anti-GOLPH3 (see below), and the mutation of the GOLPH3 gene region was confirmed by sequencing performed at Plataforma Omicas of Pontificia Universidad Católica de Chile (https://sites.google.com/bio.puc.cl/omics/inicio (accessed on 17 December 2023)). Of the four gRNA molecules used, #4 was the most efficient. MDA-MB-231 GOLPH3 KO cells were called MDA-GOLPH3-KO cells.

### 2.5. Antibodies

We used the following monoclonal antibodies: mouse anti-TOM20 (Santa Cruz Biotechnology, Dallas, TX, USA; cat # sc-17764; 1:500 dilution for immunofluorescence and 1:1000 dilution for immunoblotting), mouse anti-MFN1 (Santa Cruz Biotechnology; cat # sc-166644, 1:1000 dilution for immunoblotting), mouse anti-DRP1 (Santa Cruz Biotechnology; cat # sc-271583, 1:1000 dilution for immunoblotting), rabbit anti-phospho-DRP1 (Ser-616) (Cell Signaling Technology, Danvers, MA, USA; cat # 4494, 1:1000 dilution for immunoblotting), mouse anti-β-Actin (Santa Cruz Biotechnology; cat # sc-47778, 1:5000 dilution for immunoblotting), mouse anti-caspase 9 (Santa Cruz Biotechnology; cat # sc-56076, 1:1000 dilution for immunoblotting), mouse anti-caspase 3 (Santa Cruz Biotechnology; cat # sc-56053, 1:1000 dilution for immunoblotting), mouse anti-cytochrome c (Santa Cruz Biotechnology; cat # sc-7159, 1:1000 dilution for immunoblotting), and mouse anti-Bcl2 (Santa Cruz Biotechnology; sc-7382, 1:000 for immunoblot). We used the following polyclonal antibodies: rabbit anti-GOLPH3 (Abcam, Cambridge, UK; cat # ab98023, 1:1000 dilution for immunoblotting and 1:500 for immunofluorescence), rabbit anti-MFN1 (Santa Cruz Biotechnology; cat # sc-50330, 1:1000 dilution for immunoblotting), rabbit anti-OPA1 (Thermo Fisher Scientific, cat # PA1–16991, 1:1000 dilution for immunoblotting), and rabbit anti-MFF (Cell Signaling; cat # 84580). We used a homemade mouse polyclonal antibody against human GOLPH3 that we generated as described elsewhere [25]. We used the following secondary antibodies for immunofluorescence: Alexa Fluor-488–conjugated goat anti-rabbit IgG (Thermo Fisher Scientific; cat # A-11008, 1:500 dilution) and Alexa Fluor-555–conjugated goat anti-mouse IgG (Thermo Fisher Scientific; cat # A-21422, 1:500 dilution). We used the following secondary antibodies for immunoblotting: horseradish peroxidase (HRP)-conjugated goat anti-rabbit IgG (Thermo Fisher Scientific; cat # 31460, 1:5000 dilution) and HRP-conjugated goat anti-mouse IgG (Thermo Fisher Scientific; cat # 31430; 1:5000 dilution).

### 2.6. Immunofluorescence and Image Analysis

Cells grown on glass coverslips were washed thrice with phosphate-buffered saline (PBS) containing 0.1 mM CaCl_2_ (Sigma-Aldrich, St. Louis, MO, USA; cat # 21225-100ML) and 1 mM MgCl_2_ (Thermo Fisher Scientific; cat # AM9530G) (PBS-CM). Cells were fixed with 4% paraformaldehyde (Electron Microscopy Sciences, Hatfield, PA, USA; cat # 15710) in PBS-CM for 1 h at room temperature, followed by three washes in PBS-CM. Permeabilization was performed using 0.3% Triton X-100 in PBS-CM for 15 min at room temperature, followed by blocking with 0.5% gelatin in PBS-CM (PBS-CM-G) for 10 min at room temperature. The primary antibody anti-TOM20 was used at a 1:500 dilution in PBS-CM-G for 1 h at room temperature, followed by three washes in PBS-CM and incubation with the secondary antibody Alexa Fluor-488-conjugated goat anti-mouse IgG, at a 1:500 dilution, or Alexa Fluor-555-conjugated goat anti-mouse IgG, at a 1:500 dilution, in PBS-CM-G for 30 min at room temperature. After three washes in PBS-CM, glass coverslips were mounted on glass slides using Fluoromount-G mounting medium (Electron Microscopy Sciences; cat # 17984-25). Images were acquired in 16-bit z-stack format using a TCS SP8 laser scanning confocal microscope equipped with a PlanApo 63× oil immersion objective (NA 1.4), a 50 mW Argon laser at 488 nm, an HyD hybrid detector system, and using LASX microscopy software (version 3.5.2.18963; Leica, Wetzlar, Germany). To analyze the mitochondrial network, we used ImageJ software (version 1.53f51v; [28]) implemented with the 3D Object Counter plugin. The number of objects (equivalent to the number of mitochondria per cell) and the volume of mitochondria (in μm^3^) were quantified in a minimum of 20 cells, following a protocol described elsewhere [29]. Normalization of the mitochondrial count and volume was performed based on the area of individual cells.

### 2.7. Transmission Electron Microscopy and Quantitative Analysis of Mitochondria

Cells grown in 60 mm plates to 80% confluency were washed with PBS and fixed in 1 mL of 2.5% glutaraldehyde in 0.2 M cacodylate buffer (pH 7.2) at 4 °C. The next day, the samples were incubated in 1% aqueous OsO_4_ for 45 min and washed in double-distilled water for 10 min. The samples were incubated in a 0.5% aqueous solution of uranyl acetate for 1 h, followed by sequential washes with acetone at varying concentrations (50%, 70%, twice at 95%, and twice at 100%), with each wash lasting 10 min. Next, the samples were incubated in Epon: acetone 1:1 (Electron Microscopy Sciences; cat # EMS #14120) overnight. The samples were left in molds with resin at 60 °C for 24 h to allow the resin to polymerize and harden. The next day, to allow acetone evaporation, resin-embedded samples were left to air dry for a minimum of 8 h. The resin blocks were extracted from their molds, ultrafine cuts of 90–100 nm were obtained using a diamond knife with an Ultracut R ultramicrotome (Leica), and the resulting ultrathin sections were mounted onto copper electron microscopy grids (Electron Microscopy Sciences; cat # G200H-Cu). The samples were examined, and micrographs were obtained using a Talos F200C G2 transmission electron microscope (Thermo Fisher Scientific) operating at 80 kV. Morphometric mitochondrial analyses of micrographs were performed with Fiji software (version 2.1.0/1.53c; [30]). Intact membrane mitochondria and electron-dense mitochondria were defined as previously described [31] and are shown as a percentage of total mitochondria.

### 2.8. Preparation of Protein Extracts, SDS-PAGE, Immunoblotting, and Densitometric Quantification

Cells grown in 100 mm plates were washed thrice with ice-cold PBS and lysed at 4 °C in Triton buffer (50 mM Tris, 150 mM NaCl, 1 mM EDTA, 1% Triton X-100, pH 7.5) supplemented with a cocktail of protease inhibitors (Thermo Fisher Scientific; cat # 78429) and a cocktail of phosphatase inhibitors (1 mM NaF, 1 mM Na_2_P_2_O_7_, and 1 mM Na_3_VO_4_). Soluble extracts were centrifuged at 14,000× *g* for 20 min at 4 °C. The supernatant of each sample was collected, and protein concentration was quantified using a BCA protein assay (Thermo Fisher Scientific; cat # 23225). Proteins in samples were denatured in Laemmli Sample Buffer (Bio-Rad Laboratories, Hercules, CA, USA. cat # 1610747) for 5 min at 95 °C and resolved by SDS-PAGE followed by electrotransfer to PVDF membranes. Incubation of membranes with primary antibodies diluted in PBS supplemented with 5% non-fat milk (PBS-M) was performed overnight at 4 °C, followed by three washes at room temperature in PBS containing 0.1% Tween-20 (PBS-T). Incubation with secondary antibodies also diluted in PBS-M was performed for 1 h at room temperature, followed by three washes in PBS-T at room temperature. Immunoreactive bands were visualized in the chemiluminescence documentation system G:BOX Chemi XRQ (Syngene, Cambridge, UK) using a chemiluminescent detection reagent (Immobilon Forte Western HRP substrate, Millipore, MA, USA; cat # WBLUF0500). Densitometric quantification of immunoreactive bands was performed using ImageJ software (version 1.53f51v).

### 2.9. ATP and ROS Quantification

Total ATP content was determined using a luciferin/luciferase bioluminescence assay (Thermo Fisher Scientific; cat # A22066) normalizing the value of each sample to its protein concentration, following a method previously described [32]. To inhibit ATP synthase, cells were treated with 10 or 20 μM oligomycin (Sigma-Aldrich; cat # O4576). For quantification of mitochondrial ATP production, cells were homogenized using MSH buffer (230 mM mannitol, 70 mM sucrose, 5 mM HEPES, 1 mM EDTA, pH 7.4) supplemented with a cocktail of protease inhibitors (Thermo Fisher Scientific; cat # 78429) and a cocktail of phosphatase inhibitors (1 mM NaF, 1 mM Na_2_P_2_O_7_, and 1 mM Na_3_VO_4_). A mitochondrial fraction of cells was obtained as previously described [33]. Mitochondria-enriched fractions were incubated with the oxidative substrates 5 mM pyruvate and 2.5 mM malate, and ATP production was determined using the same luciferin/luciferase bioluminescence assay mentioned above. For ROS quantification, 25,000 cells were seeded in each well of a white 96-well SPL plate (cat # 30196) and incubated with the fluorescent dye CM-H_2_DCFDA (DCF) (Thermo Fisher Scientific; cat # C6827) at 2.5 μM for 30 min at 37 °C. Fluorescence measurements were carried out using a Synergy HTX plate multi-mode reader (Biotek, Winooski, VT, USA) following a methodology previously described [34] at an excitation wavelength of 485 nm and an emission wavelength of 530 nm. To control DCF fluorescence, cells were treated with the antioxidant Mitoquino-Mesylate (MitoQ) (MitoQ Ltd., Auckland, New Zealand; cat # 01ATP04C-02-1) for 30 min, and the fluorescence intensity was measured before and after MitoQ administration using confocal microscopy on live cells.

### 2.10. Mitochondrial Membrane Potential

Mitochondrial membrane potential was quantified from 25,000 cells seeded in black 96-well SPL plates (cat # 30296). The following day, cells were incubated with 100 nM of MitoTracker^TM^ Red CMXRos or 10 nM of TMRE for 20 min in a humidified incubator with 5% (*v*/*v*) CO_2_ at 37 °C. Subsequently, fluorescence was quantified using a Synergy HTX multi-mode reader plate (Biotek) with an excitation wavelength of 579 nm and an emission wavelength of 599 nm. The uncoupling agent carbonyl cyanide 4(trifluoromethoxy)phenylhydrazone (FCCP) (Sigma-Aldrich; cat # C2920) was used to validate mitochondrial membrane potential quantifications. Cells were treated for 30 min with 40 μM FCCP, followed by TMRE treatment and fluorescence measurements.

### 2.11. Oxygen Consumption

Oxygen consumption was quantified from 500,000 cells incubated with 5 mM pyruvate and 2.5 mM malate as oxidative substrates. Measurements were carried out using the Extracellular Oxygen Consumption Assay (Abcam; cat # ab197243), which is based on a fluorescent compound that quenches its signal upon interaction with molecular oxygen. Fluorescence measurements were carried out using the Synergy HTX multi-mode reader (Biotek) at an excitation wavelength of 380 nm and an emission wavelength of 650 nm.

### 2.12. Statistical Analysis

All data analyses were performed using GraphPad Prism (version 8.2.1; San Diego, CA, USA). Results were presented in graphs showing the mean ± standard error of the mean (SEM). Statistical significance for data comparisons was determined using Student’s *t*-test, one-way ANOVA, or two-way ANOVA, as appropriate. In the figures, statistical significance was denoted as follows: * for *p*-values < 0.05; ** for *p*-values < 0.01; and *** for *p*-values < 0.001.

## 3. Results

### 3.1. GFP-GOLPH3 Associates with the PI(4)P Reporter mCherry-PH-FAPP1 at Sites of Mitochondrial Fission

In HeLa cells, vesicles containing PI(4)P that have budded from the GA reach mitochondrial fission sites immediately before fission events [7]. Because GOLPH3 is a PI(4)P-binding protein, we hypothesized that GOLPH3 could also be associated with this type of vesicle. To test this possibility, we performed time-lapse live-cell imaging in HeLa cells expressing the PI(4)P reporter mCherry-PH-FAPP1 [35] and GFP-GOLPH3 (Figure 1A). Importantly, GFP-GOLPH3 behaves similarly to endogenous GOLPH3 in different types of cells, i.e., it associates with the GA and is contained in a large cytosolic pool [10,13]. Interestingly, we observed numerous dynamic vesicular profiles containing both mCherry-PH-FAPP1 and GFP-GOLPH3 (Figure 1B), suggesting that the association between GOLPH3 and PI(4)P extends to membrane-bound transport intermediates such as vesicles. Transfected HeLa cells were also labeled with the mitochondrial dye MitoTracker^TM^ Deep Red FM (Figure 1A). Surprisingly, we often observed vesicular profiles containing mCherry-PH-FAPP1 and GFP-GOLPH3 arriving at mitochondrial sites that subsequently underwent fission (Figure 1B). Although the frequency of these concerted events that we were able to identify was moderate (~20%), this observation suggests that GOLPH3 could be part of the mechanism driven by vesicles containing PI(4)P during the late steps of mitochondrial fission.

### 3.2. GFP-GOLPH3 Arrives at Mitochondrial Fission Sites after the Recruitment of YFP-DRP1

To investigate further whether GOLPH3 participates in the mechanism of mitochondrial fission driven by vesicles containing PI(4)P, we also performed time-lapse live-cell imaging, but in HeLa cells co-expressing GFP-GOLPH3 and the mitochondrial fission protein DRP1 fused to YFP (YFP-DRP1; [36]), with mitochondria also labeled with MitoTracker^TM^ Deep Red FM (Figure 2A). As expected, YFP-DRP1 was observed throughout the mitochondrial network at sites that underwent fission (Figure 2A,B). Surprisingly, HeLa cells exhibited vesicular profiles containing GFP-GOLPH3 that were eventually associated with mitochondrial sites already marked with YFP-DRP1 and that later underwent fission (Figure 2B). Specifically, we often observed that at the initial phase of a mitochondrial fission event, YFP-DRP1 was recruited to a mitochondrial site devoid of GFP-GOLPH3 (Figure 2B, 2:58.6). A few seconds later, a vesicular profile containing GFP-GOLPH3 localized at the fission site at the vicinity of YFP-DRP1 (Figure 2B, 2:59.5), followed by their colocalization during ~1–4 s (Figure 2B, 3:02.9–3:04.7). Notably, after mitochondrial fission occurred (Figure 2B, 3:04.7), we observed dissociation of GFP-GOLPH3 and YFP-DRP1 (Figure 2B, 3:06.4), followed by apparent mitochondrial detachment and dispersal of both the GFP-GOLPH3 vesicular profile and YFP-DRP1, and this occurred concomitantly with consolidated mitochondrial fission (Figure 2B, 3:10.7–3:15.1). We analyzed 85 time-lapse live-cell imaging videos and found 184 mitochondrial fission events, of which we were able to identify 38 of them (~21%) that featured the recruitment of GFP-GOLPH3. These data suggest that GOLPH3 is recruited to the mitochondrial network to participate in at least a fraction of mitochondrial fission events.

Because the overexpression of GOLPH3 in MDA-MB-231 breast cancer cells has been implicated in perturbations of mitochondrial metabolism [27], we wondered whether the recruitment of GOLPH3 could also happen at the mitochondrial fission sites of these cells. To test this, we co-expressed in these cells GFP-GOLPH3 and YFP-DRP1 and labeled mitochondria with MitoTracker^TM^ Deep Red FM (Figure 3A). In addition to the perinuclear distribution of GFP-GOLPH3, we observed a large fluorescent signal in the cytoplasm of MDA-MB-231 cells, which was larger compared with HeLa cells (Figure 3A). This was, however, expected because, compared with other cell lines, MDA-MB-231 cells contain a larger cytosolic pool of endogenous GOLPH3, as well as of exogenously expressed GFP-GOLPH3 [10]. Similar to HeLa cells, we observed numerous YFP-DRP1 puncta associated with the mitochondrial network of MDA-MB-231 cells (Figure 3A). Importantly, we also observed vesicular profiles containing GFP-GOLPH3 that were recruited at mitochondrial fission sites marked by YFP-DRP1 (Figure 3B, 0:53.5–1:15.9). Similarly, after mitochondrial fission occurred (Figure 3B, 1:15.9), we also observed dissociation of GFP-GOLPH3 and YFP-DRP1 (Figure 3B, 2:16.4), followed by apparent mitochondrial detachment and dispersal of both GFP-GOLPH3 vesicular profiles and YFP-DRP1 (Figure 3B, 3:01.2). Thus, our observations suggest that in different cell types, GOLPH3 is recruited to the mitochondrial network to participate in at least a fraction of mitochondrial fission events.

### 3.3. The Depletion of GOLPH3 in MDA-MB-231 Cells Disrupts the Structure of the Mitochondrial Network

Mitochondria are dynamic organelles that undergo highly coordinated fission and fusion events [36]. To determine whether the presence of GOLPH3 at sites of mitochondrial fission plays a role in the dynamic structure of the mitochondrial network, we analyzed the effect of the disruption of GOLPH3 expression in MDA-MB-231 cells. We used CRISPR-Cas9 gene editing to generate GOLPH3 KO MDA-MB-231 cells (MDA-GOLPH3-KO cells). We confirmed the depletion of GOLPH3 expression by immunoblot and immunofluorescence analysis (Appendix A). Next, to determine the effect of the depletion of GOLPH3 expression on the structure of the mitochondrial network, we performed indirect immunofluorescence using an antibody against TOM20, a mitochondrial outer membrane protein [37]. In wild-type MDA-MB-231 (MDA-WT) cells, we observed an interconnected tubular mitochondrial network (Figure 4A and Appendix A), whereas in MDA-GOLPH3-KO cells, we observed a fragmented and less interconnected mitochondrial network (Figure 4A and Appendix A). Quantitative analysis of the mitochondrial network revealed a significant increase in the number of mitochondrial profiles in MDA-GOLPH3-KO cells to 113.7 ± 5.1% compared to MDA-WT cells (Figure 4B), an indicator of mitochondrial fragmentation [29]. Additionally, our analysis showed a significant reduction in the volume of mitochondrial profiles in MDA-GOLPH3-KO cells to 40.6 ± 5.7% compared with MDA-WT cells (Figure 4C). These results indicate that the depletion of GOLPH3 expression in MDA-MB-231 cells produces mitochondrial network fragmentation and further suggests that GOLPH3 is implicated in the mitochondrial fission process.

Changes in mitochondrial fusion and fission correlate with changes in the structural integrity of mitochondrial membranes [37,38]. Thus, we decided to perform a quantitative analysis of the structural integrity of mitochondrial membranes by transmission electron microscopy. The analysis showed that most mitochondria in MDA-WT cells had mitochondrial cristae with intact membranes (Figure 5A,B). In contrast, in the mitochondria of MDA-GOLPH3-KO cells, we found a significant loss of the integrity of the mitochondrial outer membrane to 62.0 ± 38.1% compared with MDA-WT cells (Figure 5A, orange arrows, and B). Loss of mitochondrial membrane integrity often correlates with decreased electron density observed in electron microscopy images, which is a parameter often correlated with the functional state of mitochondria [39]. Thus, we compared the electron density of mitochondria; however, we found no significant difference between MDA-WT and MDA-GOLPH3-KO cells (Figure 5C). Nevertheless, our data indicate that the depletion of GOLPH3 expression in MDA-MB-231 cells produces mitochondrial membrane disruption, suggesting that GOLPH3 is necessary for the structural and functional integrity of mitochondria.

### 3.4. The Depletion of GOLPH3 in MDA-MB-231 Cells Affects the Levels of Mitochondrial Fusion and Fission Proteins

The mitochondrial fragmentation observed in MDA-GOLPH3-KO cells suggested an imbalance in the mitochondrial fusion and fission processes, which is likely due to a decrease in the mitochondrial fusion rate. Thus, we next analyzed by immunoblot whether the depletion of GOLPH3 expression in MDA-MB-231 cells modified the levels of proteins associated with mitochondrial fusion and fission. First, we analyzed the levels of MFN1, MFN2, and OPA1, three proteins involved in mitochondrial fusion [40]. We found a significant decrease in the levels of MFN1 and OPA1 in MDA-GOLPH3-KO cells to 81.1 ± 4.6% and 64.2 ± 9.3%, respectively, compared with MDA-WT cells (Figure 6A,B). In contrast, we found no significant differences in MFN2 levels (Figure 6A,B). Nevertheless, the reduced levels of MFN1 and OPA1 suggest that GOLPH3 is implicated in the expression of these mitochondrial fusion proteins. Next, we analyzed the levels of total DRP1, as well as of phosphorylated DRP1 at serine 616 (pDRP1), which is a specific phosphorylation that allows the translocation of DRP1 from the cytosol to the outer mitochondrial membrane, promoting mitochondrial fission [41]. We found a significant increase in the total levels of DRP1 in MDA-GOLPH3-KO cells to 177.2 ± 31.6% compared with MDA-WT cells (Figure 6C,D). Unexpectedly, we found a significant decrease in the levels of pDRP1 in MDA-GOLPH3-KO cells to 53.9 ± 5.6% compared with MDA-WT cells (Figure 6C,D), suggesting that phosphorylation of DRP1 at serine 616 is largely impaired upon GOLPH3 depletion not being compensated with the increased levels of DRP1. We also analyzed the levels of MFF, a protein with a prominent role in mitochondrial fission [42] essential for mitochondrial recruitment of DRP1 during mitochondrial fission in mammalian cells [43]. We found a significant reduction in the levels of MFF in MDA-GOLPH3-KO cells to 72.3 ± 11.8% compared with MDA-WT cells (Figure 6C,D). These results suggest that GOLPH3 is necessary to regulate the expression of key proteins involved in mitochondrial fission.

### 3.5. The Depletion of GOLPH3 in MDA-MB-231 Cells Reduces the Mitochondrial Bioenergetic Capacity

It has been shown that changes in mitochondrial mass affect the mitochondrial bioenergetic function, potentially impacting ATP production [44]. It has also been shown that under physiological conditions, the distribution of mitochondria throughout the cell and mitochondrial fusion and fission events are related to mitochondria metabolic activity [45]. Because we observed in MDA-GOLPH3-KO cells a fragmented mitochondrial network with loss of integrity in the mitochondrial outer membrane, we decided to analyze whether this change in morphology correlated with an impact on the mitochondrial bioenergetic function. We first quantified the total levels of ATP (ATP content) in cell lysates using a luminescent assay. We found in MDA-GOLPH3-KO cells a significant decrease in ATP levels to 73.7 ± 5.4% compared with MDA-WT cells (Figure 7A). In addition, we quantified ATP levels in the lysate of cells treated with the ATP synthase inhibitor oligomycin [46]. We found that inhibiting ATP synthase significantly reduced ATP content in MDA-GOLPH3-KO cells compared with MDA-WT cells (Appendix A), suggesting an impaired ATP production capacity in the mitochondria of cells depleted of GOLPH3. To test this possibility, we compared ATP production in mitochondria-enriched subcellular fractions supplemented with the oxidative substrates malate and pyruvate. We observed that the mitochondria-enriched fraction of MDA-GOLPH3-KO cells exhibited a significant decrease in ATP production to 73.6 ± 4.6% compared with MDA-WT cells (Figure 7B), indicating that GOLPH3 depletion impairs mitochondrial ATP production in MDA-MB-231 cells. To explore whether the decrease in ATP levels and mitochondrial ATP production capacity in MDA-GOLPH3-KO cells was due to impaired respiratory function, we analyzed some factors necessary for ATP production during oxidative phosphorylation. Because the mitochondrial membrane potential generated by proton pumps is an essential element during oxidative phosphorylation [47], we analyzed mitochondrial membrane potential by incubating cells with either of the fluorescent mitochondrial dyes MitoTracker^TM^ Red CMXRos or TMRE and quantified the respective levels of fluorescence. In cells incubated with MitoTracker^TM^ Red CMXRos, we found significantly lower fluorescence levels in MDA-GOLPH3-KO cells to 24.4 ± 1.8% compared with MDA-WT cells (Figure 7C). Similarly, we found a significantly lower fluorescence in MDA-GOLPH3-KO cells incubated with TMRE to 63.3 ± 10.1% compared with MDA-WT cells (Figure 7D). To validate that the quantified fluorescence was a measure of mitochondrial membrane potential, we used the uncoupling agent carbonyl cyanide 4(trifluoromethoxy)phenylhydrazone (FCCP) [48]. Cells were treated for 30 min with 40 μM FCCP, followed by incubation with TMRE. Treatment with FCCP resulted in a significant decrease in the levels of fluorescence in both MDA-WT and MDA-GOLPH3-KO cells, which is indicative of a reduction in the mitochondrial membrane potential after mitochondrial depolarization compared with untreated cells (Appendix A). These data indicate that the depletion of GOLPH3 expression in MDA-MB-231 cells reduced the mitochondrial membrane potential. Considering that oxidative phosphorylation uses oxygen as the final oxidizing agent [49], we next analyzed whether there was also a decrease in oxygen consumption in MDA-GOLPH3-KO cells. We incubated cells with the oxidative substrates pyruvate and malate. We measured extracellular oxygen consumption rates using a fluorescent compound that is quenched in the presence of oxygen. We found a significant decrease in the fluorescence levels of MDA-GOLPH3-KO cells to 59.4 ± 14.0% compared with MDA-WT cells (Figure 7E), indicative of higher levels of fluorescence-quenching oxygen, which suggests a decrease in oxygen consumption upon depletion of GOLPH3 expression in MDA-MB-231 cells likely by impaired oxidative phosphorylation. When oxidative phosphorylation is impaired, unpaired electrons can escape and react with oxygen, increasing the levels of ROS [50,51]. Thus, we analyzed whether the depletion of GOLPH3 expression impacted ROS content. Cells were incubated with CM-H2DCFDA (DCF), a non-fluorescent compound that, in the presence of ROS, is rapidly oxidized, emitting fluorescence. We found a significant increase in ROS in MDA-GOLPH3-KO cells to 160.5 ± 16.8% compared with MDA-WT cells (Figure 7F). To demonstrate that the levels of DCF fluorescence were a measure of ROS content, we treated cells for 30 min with MitoQ, a mitochondria-targeted antioxidant [52], and measured the fluorescence intensity before and after MitoQ administration using confocal microscopy on live cells. We observed a decrease in fluorescence intensity (Appendix A), which is indicative of a reduction in reactive oxygen species due to its antioxidant capacity. These data suggest that the depletion of GOLPH3 expression in MDA-MB-231 cells generated an oxidative imbalance that led to elevated ROS levels, possibly due to increased ROS production in the mitochondria or decreased cellular antioxidant controls. Finally, considering that mitochondrial dysfunction induces apoptosis [53], we quantified by immunoblot analysis the levels of important apoptotic mediators, including caspase 9, caspase 3, and cytochrome-c, and the anti-apoptotic factor Bcl-2 [53]. We found similar levels of all these proteins in both MDA-GOLPH3-WT and MDA-GOLPH3-KO cells (Appendix A), indicating that the depletion of GOLPH3 did not induce programmed cell death. Altogether, these data indicate that MDA-GOLPH3-KO cells have mitochondrial dysfunction, evidenced by reduced mitochondrial ATP production, a loss of mitochondrial membrane potential, decreased oxygen consumption, and increased ROS content without apoptosis induction, suggesting that the expression of GOLPH3 plays a role in mitochondrial bioenergetics.

## 4. Discussion

It has been proposed that vesicles containing PI(4)P budded off the GA of HeLa cells could function in mitochondrial fission [7]. These vesicles localize to fission sites after the arrival of DRP1, suggesting a function of these vesicles in the last stage of mitochondrial fission [7]. Moreover, at the contact sites between these vesicles and mitochondria, actin polymerizes, promoting mitochondrial fission, and an increase in vesicles budded off the GA containing PI(4)P increases actin polymerization and reduces mitochondrial length [8]. These findings suggest a complex inter-organelle communication mechanism between the GA and mitochondria to sustain the dynamic structure of the mitochondrial network. However, the role that PI(4)P might play in mitochondrial fission is unknown. One possibility is that PI(4)P could serve to bind effectors contributing to this process. We propose that the PI(4)P-binding protein GOLPH3 is a candidate. Using live-cell imaging, we evaluated in HeLa cells whether GFP-GOLPH3 was localized in PI(4)P-containing vesicles at mitochondrial fission sites. We used the PI(4)P reporter mCherry-PH-FAPP1 and found that it colocalized with GFP-GOLPH3 in vesicular profiles, suggesting that in addition to the GA, GOLPH3 is also recruited to PI(4)P-containing vesicles. Moreover, we found that vesicular profiles containing mCherry-PH-FAPP1 and GFP-GOLPH3 eventually localized at mitochondrial fission sites. To determine the relationship between the GFP-GOLPH3-containing vesicular profiles and the mitochondrial fission machinery, we performed live-cell imaging in HeLa cells to detect the mitochondrial fission protein DRP1 using YFP-DRP1 as a reporter. Consistent with the findings that PI(4)P-containing vesicles arrive at mitochondrial fission sites after the recruitment of DRP1 [7], we found that GFP-GOLPH3-containing vesicular profiles localized at these sites after the recruitment of YFP-DRP1. Because GOLPH3 behaves differently in different cell lines, such as in the breast cancer cell lines MCF-7 and MDA-MB-231 [10], and because the overexpression of GOLPH3 in MDA-MB-231 affects mitochondrial function [27], we evaluated whether GFP-GOLPH3 was also found at mitochondrial fission sites in MDA-MB-231 cells. Thus, our finding of similar recruitment of GFP-GOLPH3 at mitochondrial fission sites in different types of cells strongly suggests that it is a conserved function. Nevertheless, we did not detect GFP-GOLPH3 in all mitochondrial fission events, suggesting that the function of GOLPH3 could be important in the fission process under specific conditions or for specific pools of mitochondria. Alternatively, our live-cell imaging analysis using GFP-GOLPH3 could have been not resolutive enough to account for the presence of GOLPH3 in all mitochondrial events. What could be the function of GOLPH3 during mitochondrial fission? It is known that actin filaments are also involved in this process, helping the constriction of mitochondria in conjunction with DRP1 [8,54]. On the other hand, it has been shown that GOLPH3 interacts with the unconventional myosin MYO18A to mediate the mechanism necessary for functional GA perinuclear organization that also requires actin filaments [55]. Thus, GOLPH3 at mitochondrial fission sites could mediate a related mechanism between actin filaments and mitochondria, perhaps binding some type of myosin, such as the unconventional myosin MYO19 [56].

To obtain more insight into the possible functional significance of the presence of GOLPH3 at mitochondrial fission sites, we evaluated the impact of GOLPH3 depletion in MDA-MB-231 cells. If the function of GOLPH3 were to promote mitochondrial tubule constriction during fission, we would expect to find that GOLPH3 depletion would have caused a more interconnected mitochondrial network. However, we found that the knockout of GOLPH3 resulted in a decrease in the volume and an increase in the number of mitochondria, which is explained either by less mitochondrial fusion, greater mitochondrial fission, or both. It has been shown that an increase in mitochondrial fragmentation could be mainly due to a decrease in the rate of mitochondrial fusion [57]. Consistent with this possibility, we found that the depletion of GOLPH3 reduced the levels of the mitochondrial fusion proteins MFN1 and OPA1. On the other hand, we found increased levels of the mitochondrial fission protein DRP1 upon GOLPH3 depletion, which is consistent with the possibility that increased fission could also have contributed to the observed mitochondrial network fragmentation. We also found decreased levels of phosphorylated DRP1 at serine 616 upon GOLPH3 depletion. This is intriguing because this is an important phosphorylation for the translocation of DRP1 from the cytosol to the mitochondrial outer membrane to exert its function in mitochondrial fission [58]. However, it has been described that DRP1 can be regulated not only by phosphorylation but also by other post-translational modifications, such as S-nitrosylation [59]. Thus, upon GOLPH3 depletion, the higher levels of DRP1, if functional, might have exerted its function in mitochondrial fission through another type of regulation. Furthermore, mitochondrial fission could occur independently of DRP1 and its phosphorylation status. For instance, the epidermal growth factor receptor (EGFR), after binding to its ligand, can translocate to the outer mitochondrial membrane to regulate mitochondrial fission by interacting with MFN1, intervening with its polymerization state that is important for MFN1 function [60]. In this regard, it has been shown that GOLPH3 regulates the intracellular trafficking of EGFR [61]. Thus, GOLPH3 depletion in MDA-MB-231 cells may have altered EGFR trafficking in a way that contributed to a DRP1-independent mitochondrial fission mechanism. Nevertheless, we also found that the depletion of GOLPH3 expression resulted in decreased levels of MFF, which is an essential protein necessary for the mitochondrial recruitment of DRP1 [43]. This result predicted an opposite scenario to that observed in the mitochondrial network upon GOLPH3 depletion, i.e., a more interconnected mitochondrial network. Therefore, our data suggest that the expected effects of reduced MFF levels in GOLPH3-depleted cells were not strong enough to affect mitochondrial fission. Thus, collectively, our data suggest that the expression of GOLPH3 is implicated in an intricate mechanism that regulates the levels of at least a set of mitochondrial fusion and fission proteins, strongly implicating GOLPH3 as an important player in the functional inter-organelle communication between the GA and mitochondria. How could the observation of GFP-GOLPH3 at mitochondrial fission sites be reconciled with increased fragmentation in cells depleted of GOLPH3? We propose that GOLPH3 could be a negative regulator of mitochondrial fission.

Because the knockdown of GOLPH3 leads to a decrease in mitochondrial mass [26], and because we found that the depletion of GOLPH3 in MDA-MB-231 cells resulted in a more fragmented mitochondrial network, as well as a more disrupted outer mitochondrial membrane, we assessed whether GOLPH3 depletion in these cells resulted in functional effects such as on mitochondrial bioenergetics, and we found decreased mitochondrial membrane potential upon GOLPH3 depletion. Consistent with the fact that the mitochondrial membrane potential is an essential component for productive oxidative phosphorylation [47], we found that the depletion of GOLPH3 resulted in a decrease in the total levels of ATP, as well as in mitochondrial ATP production, suggesting a putative mitochondrial dysfunction. How could GOLPH3 mediate a negative regulation in mitochondrial fission? Interestingly, it has been shown that GOLPH3 of human glioma U251 cells interacts with the cytosolic proteins PHB1 and PHB2 [62], which have also been observed to be associated with mitochondria [63]. Moreover, the silencing of GOLPH3 reduces the levels of these proteins [62]. Importantly, PHB1 and PHB2 have been linked to mitochondrial function [63]. Depletion of PHB2 in MEF and HeLa cells leads to a decrease in the levels of long isoforms of OPA1 (a protein related to mitochondrial inner membrane fusion), resulting in a fragmented and disorganized mitochondrial network [64,65], similar to what we found upon GOLPH3 depletion. In addition, PHB1 and PHB2 have a role in oxidative phosphorylation, where the silencing of PHB1 in endothelial cells has been shown to reduce Complex I electron transport activity, increasing ROS levels [66]. Because the mitochondrial recruitment mechanism of PHB1 and PHB2 is unknown, an intriguing possibility is that their interaction with GOLPH3 could facilitate their mitochondrial function, explaining the functional effect of GOLPH3 depletion.

GOLPH3 is increasingly recognized for its role in protein and lipid glycosylation within the early secretory pathway [67,68], as well as in post-Golgi protein trafficking [69]. Similarly, experimental evidence strongly supports the tumorigenic potential of GOLPH3 overexpression and the anti-tumor effects of GOLPH3 silencing in various in vivo systems [15,19], including breast cancer [70] and glioma [71]. Remarkably, combinatorial anti-tumor strategies employing GOLPH3 knockdown have already been proven effective in glioma [72,73], colon cancer [74,75], and oral squamous cell carcinoma [76]. Meanwhile, the development of therapies targeting triple-negative breast cancer cells, a heterogeneous group of breast carcinomas, remains challenging [77]. Targeted therapies disrupting mitochondrial function and inhibiting GOLPH3 could significantly affect the secretory pathway and potentially enhance an anti-tumor immune response. For instance, such treatments may alter the surface levels and functionality of the immune checkpoint protein PD-L1. Changes in the cellular redox state and energy balance can influence PD-L1 folding and trafficking in the endoplasmic reticulum [78]. Additionally, inhibiting GOLPH3 might impact the post-Golgi trafficking and glycosylation of PD-L1, which is essential for its binding to PD-1 and subsequent immunosuppressive activity [79]. Therefore, targeting GOLPH3 may reduce mitochondrial oxygen consumption, a key factor in tumor hypoxia [80], as well as reduce PD-L1 functionality. In this regard, cancer treatments such as photodynamic therapy or radiation therapy have been shown to benefit from disrupting both oxidative phosphorylation and the function of the immune checkpoint proteins PD-1 and PD-L1, as well as inducing tumor cell death [81,82]. This could reduce the inhibitory signal mediated by PD-1, potentially reinvigorating the immune response against the tumor.

In conclusion, our results show that in HeLa and MDA-MB-231 cells, GFP-GOLPH3 is recruited to YFP-DRP1-marked mitochondrial fission sites in PI(4)P-containing vesicles. On the other hand, the depletion of GOLPH3 in MDA-MB-231 cells altered the mitochondrial network organization, resulting in mitochondrial fragmentation characterized by a network with reduced volume and an increased number of mitochondria. Furthermore, the depletion of GOLPH3 expression resulted in a decrease in mitochondrial bioenergetic function, with adverse effects on ATP production and oxygen consumption and an increase in ROS generation. We propose that GOLPH3 is a negative regulator of mitochondrial fission with a functional impact on mitochondrial bioenergetics.

## Figures and Tables

**Figure 1 cells-13-00316-f001:**
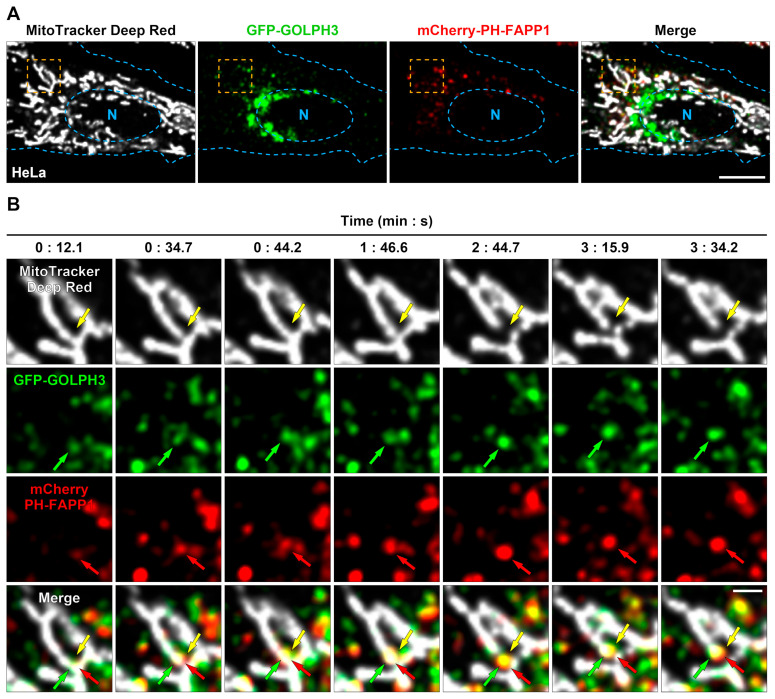
GFP-GOLPH3 and mCherry-PH-FAPP1 colocalize in mitochondrial fission sites. HeLa cells grown in 35 mm glass-bottom culture dishes were co-transfected to transiently express GFP-GOLPH3 and the PI(4)P reporter mCherry-PH-FAPP1. After 16 h, cells were incubated with MitoTracker^TM^ Deep Red FM for 20 min at 37 °C, followed by the transfer of culture dishes to a microscope heating stage for time-lapse live-cell imaging. Images were acquired every ~0.87 s. (**A**) Representative imaged cell showing the cytoplasmic distribution of the labeled mitochondrial network (MitoTracker Deep Red; gray channel), GFP-GOLPH3 (green channel), and mCherry-PH-FAPP1 (red channel). The fourth image depicts the superposition of the gray, green, and red channels (Merge). The boundary of the cell and nucleus (N) is marked with pale blue dashed lines. Bar, 10 μm. (**B**) Magnification of time-lapse images acquired at the indicated times of the cell region highlighted with an orange dashed line in the images shown in (**A**). The yellow arrows depict a mitochondrial fission event; the green arrows depict the recruitment of a GFP-GOLPH3 vesicle at the site of the mitochondrial fission highlighted by the yellow arrow; and the red arrows depict the presence of mCherry-PH-FAPP1 in the same vesicle that contains GFP-GOLPH3. Bar, 2 μm.

**Figure 2 cells-13-00316-f002:**
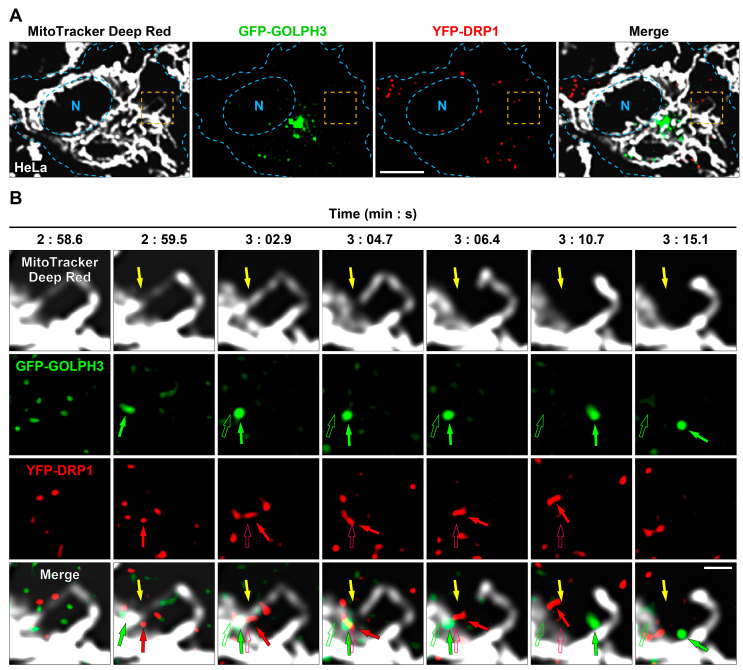
GFP-GOLPH3 localizes to mitochondrial fission sites after the recruitment of YFP-DRP1. HeLa cells grown in 35 mm glass-bottom culture dishes were co-transfected to express GFP-GOLPH3 and the mitochondrial fission protein YFP-DRP1. After 16 h, cells were incubated with MitoTracker^TM^ Deep Red FM for 20 min at 37 °C, followed by the transfer of culture dishes to a microscope heating stage for time-lapse live-cell imaging. Images were acquired every ~0.87 s. (**A**) Representative imaged cell showing the cytoplasmic distribution of the labeled mitochondrial network (MitoTracker Deep Red; gray channel), GFP-GOLPH3 (green channel), and YFP-DRP1 (red channel). The fourth image depicts the superposition of the gray, green, and red channels (Merge). The boundary of the cell and nucleus (N) is marked with pale blue dashed lines. Bar, 10 μm. (**B**) Magnification of time-lapse images acquired at the indicated times of the cell region highlighted with an orange dashed line in the images shown in (**A**). The yellow arrows depict a mitochondrial fission event; the green solid arrows depict the recruitment and displacement of a GFP-GOLPH3 vesicle at the site of the mitochondrial fission highlighted by yellow arrows; and the red solid arrows depict the recruitment and displacement of YFP-DRP1 at the site of the mitochondrial fission highlighted by the yellow arrow. The green and red open arrows depict the initial position of the respective solid arrows. Bar, 2 μm.

**Figure 3 cells-13-00316-f003:**
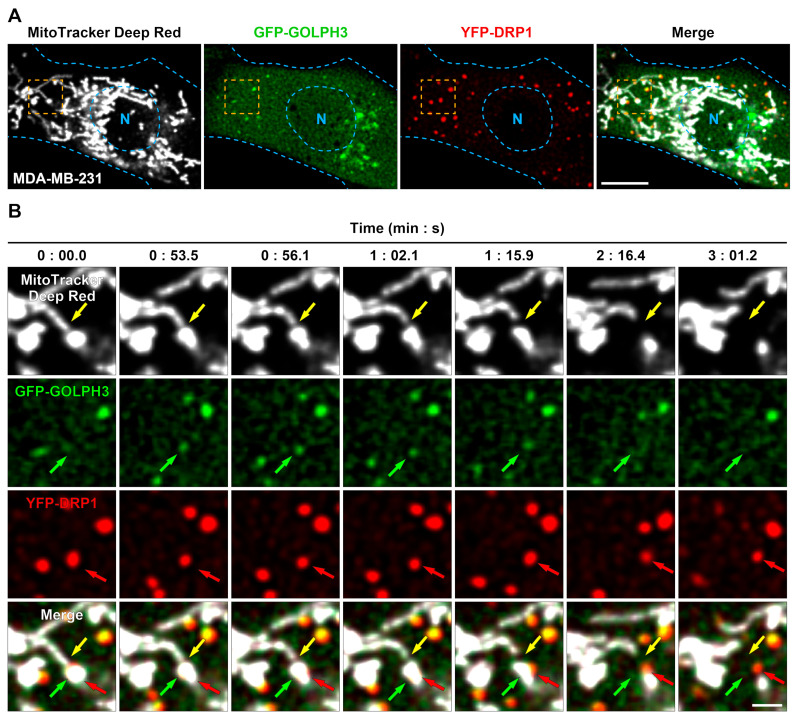
GFP-GOLPH3 localizes to mitochondrial fission sites after the recruitment of YFP-DRP1 in MDA-MB-231 cells. MDA-MB-231 cells grown in 35 mm glass-bottom culture dishes were co-transfected to express GFP-GOLPH3 and YFP-DRP1. After 16 h, cells were incubated with MitoTracker^TM^ Deep Red FM for 20 min at 37 °C, followed by the transfer of culture dishes to a microscope heating stage for time-lapse live-cell imaging. Images were acquired every ~0.86 s. (**A**) Representative imaged cell showing the cytoplasmic distribution of the labeled mitochondrial network (MitoTracker Deep Red; gray channel), GFP-GOLPH3 (green channel), and YFP-DRP1 (red channel). The fourth image depicts the superposition of the gray, green, and red channels (Merge). The boundary of the cell and nucleus (N) is marked with pale blue dashed lines. Bar, 10 μm. (**B**) Magnification of time-lapse images acquired at the indicated times of the cell region highlighted with an orange dashed line in the images shown in (**A**). The yellow arrows depict a mitochondrial fission event; the green arrows depict the recruitment and displacement of a GFP-GOLPH3 vesicle at the site of the mitochondrial fission highlighted by yellow arrows; and the red arrows depict the recruitment and displacement of YFP-DRP1 at the site of the mitochondrial fission highlighted by yellow arrows. Bar, 2 μm.

**Figure 4 cells-13-00316-f004:**
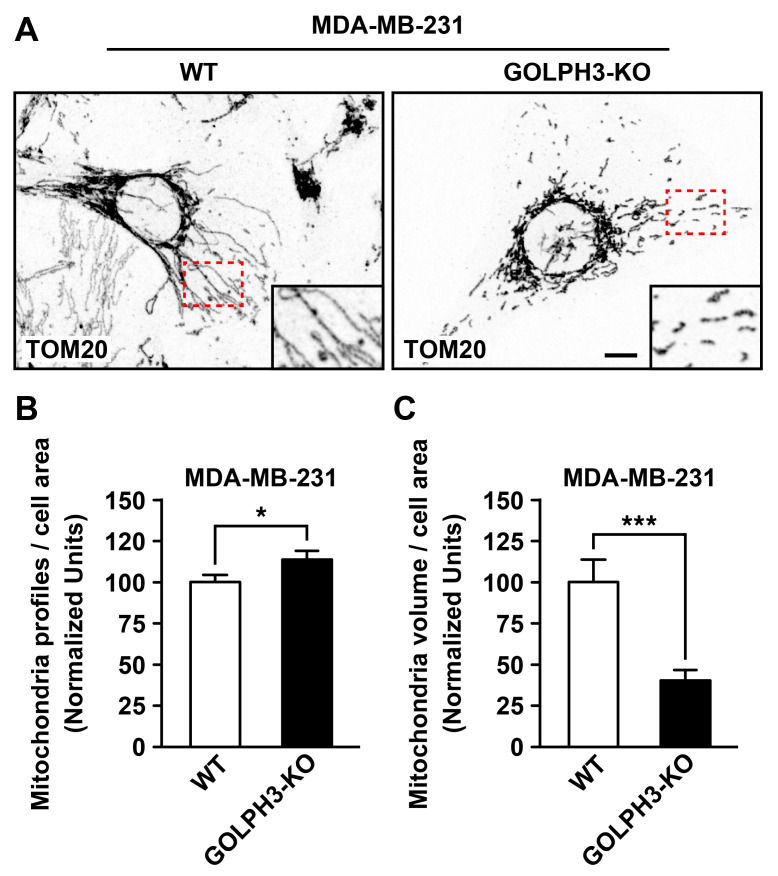
The depletion of GOLPH3 in MDA-MB-231 cells disrupts the mitochondrial network. (**A**) The indicated cells grown on glass coverslips were fixed, permeabilized, and decorated with mouse monoclonal antibodies against the mitochondrial protein TOM20. Stained cells were examined by fluorescence microscopy, and images were obtained in z-stack. The inset at the lower right corner of each image is a magnification of the respective region highlighted with a red dashed box. Bar, 10 μm. (**B**,**C**) Quantification of the number of mitochondrial profiles per cell area (**B**) and mitochondrial volume per cell area (**C**) performed using the 3D counter plugin of the ImageJ program. Bars represent the mean ± SEM (n = 3 independent experiments; n_1_ = 15 cells, n_2_ = 15 cells, n_3_ = 26 cells). * *p* < 0.05; *** *p* < 0.001.

**Figure 5 cells-13-00316-f005:**
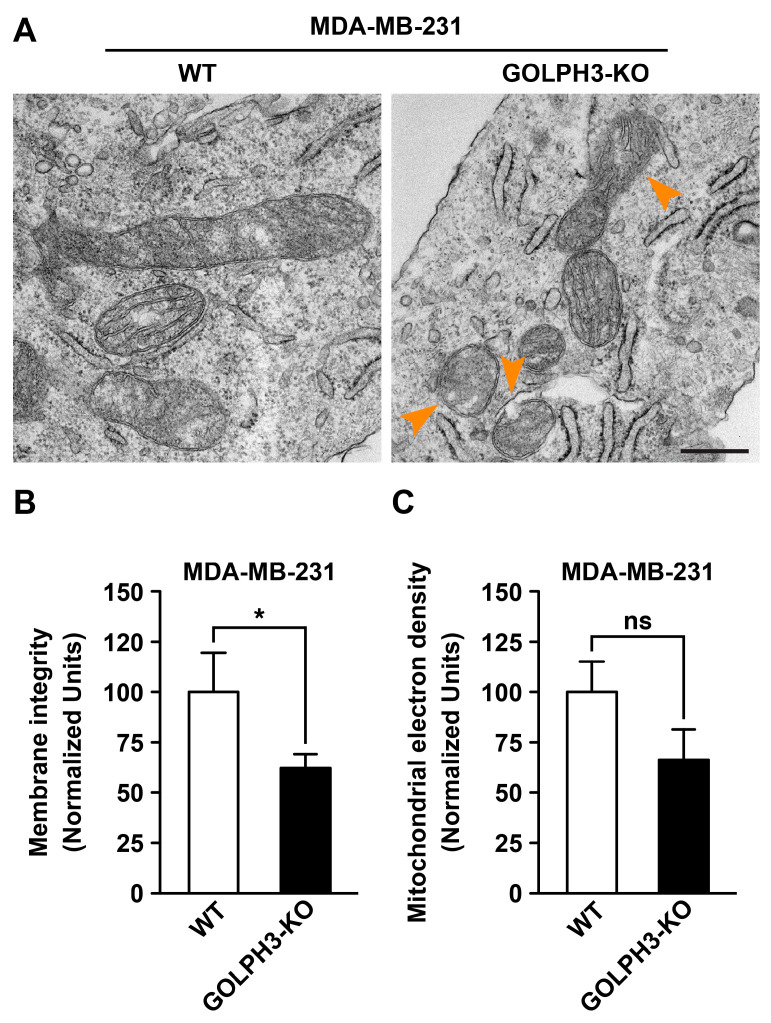
The depletion of GOLPH3 in MDA-MB-231 cells disrupts the outer mitochondrial membrane. The indicated cells were processed for transmission electron microscopy, and the images obtained (**A**) were analyzed to quantify outer mitochondrial membrane integrity (**B**) and electron density (**C**) using morphometric mitochondrial analyses implemented in Fiji software (version 2.1.0/1.53c). The orange arrow heads depict loss of the integrity of the mitochondrial outer membrane. Bar, 500 μm. Bars represent the mean ± SEM (n = 3 independent experiments; n_1_ = 6 cells, n_2_ = 6 cells, n_3_ = 6 cells). * *p* < 0.05; ns: not statistically significant.

**Figure 6 cells-13-00316-f006:**
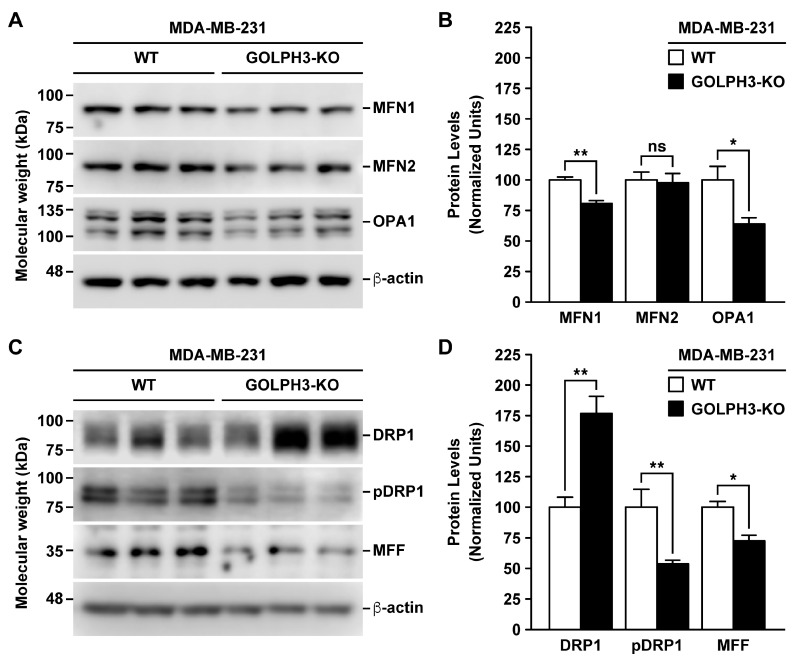
The depletion of GOLPH3 in MDA-MB-231 cells modifies the levels of mitochondrial fusion and fission proteins. (**A**,**C**) Detergent-soluble extracts were prepared from the indicated cells grown in 6-well plates, and the samples were processed by SDS-PAGE and immunoblot analysis using antibodies to detect the proteins indicated on the right involved in mitochondrial fusion (**A**) or mitochondrial fission (**C**). The immunoblot signal of anti-β-actin was used as the loading control. The position of molecular mass markers is indicated on the left. (**B**,**D**) Quantification of the respective immunoblot signal as shown in A and C. In C and D, pDRP1 denotes DRP1 phosphorylated at serine 616. Bars represent the mean ± SEM (n = 3). * *p* < 0.05; ** *p* < 0.01; ns: not statistically significant.

**Figure 7 cells-13-00316-f007:**
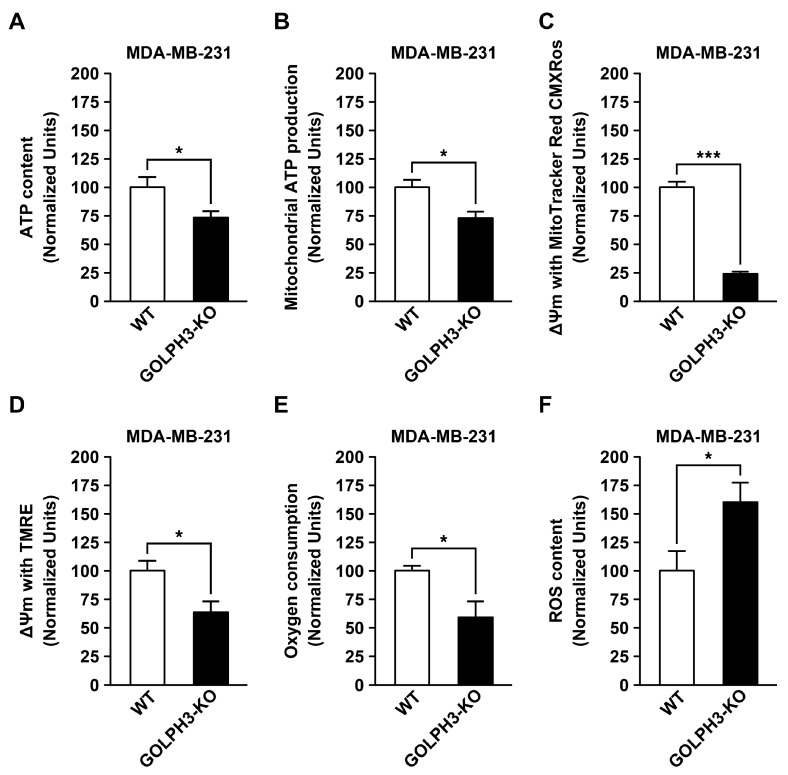
GOLPH3 depletion in MDA-MB-231 cells results in decreased mitochondrial bioenergetics capacity and increased ROS content. (**A**) Protein extracts were prepared from the indicated cells, and ATP content was quantified using a luminescence assay (n = 15). (**B**) ATP production in a mitochondria-enriched fraction from the indicated cells incubated with oxidative substrates was quantified using a luminescence assay as in (**A**) (n = 5). (**C**,**D**) Mitochondrial membrane potential was quantified by incubating the indicated cells with the fluorescent dyes MitoTracker^TM^ Red CMXRos (n = 3) (C) or with TMRE (n = 3) (**D**). (**E**) Oxygen consumption was quantified in the indicated cells using a fluorescent assay of cells incubated with an oxygen-quenchable fluorescent compound (n = 5). (**F**) ROS levels were quantified in the indicated cells using the fluorescent dye CM-H2DCFDA (n = 4). Bars represent the mean ± SEM. * *p* < 0.05; *** *p* < 0.001.

## Data Availability

The original contributions presented in the study are included in the article and Appendix A, further inquiries can be directed to the corresponding authors.

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
