# Peer review of "GOLPH3 Participates in Mitochondrial Fission and Is Necessary to Sustain Bioenergetic Function in MDA-MB-231 Breast Cancer Cells"

_cells, 2024, doi:10.3390/cells13040316_

Round 1
Reviewer 1 Report
Comments and Suggestions for Authors
Catalina M. Polanco and colleagues’s work is focused on GOLPH3, a PI(4)P-interacting GA protein, that could be involve in interorganelle communication between golgi and mitochondria. They have analyse the impact of GOLPH3 overexpression in HelA cells and GOLPH3 depletion in a breast cancer cell line, MDA-MB-231 cells. They have particularly focused on the impact of GOLPH3 depletion in MDA-MB-231 cells. Thus they have demontrate that GOLPH3 not only affect mitochondrial morphology but also impact ATP production and mitochondrial function. This work is particularly interesting but some point should be clarified. The fluorescent background observed with GFP-GOLPHE3 in MDA-MB-231 cells is quite high. The authors should give some explanation or should present picture with a lower exposition time. Fig 4 and fig 5 the legend should be completed. Indeed, the authors should explain their quantification: how much cells were counted from how much independant experiments? To clearly compare global ATP level with mitochondrial ATP it should be more relevant to measured the global ATP level in cell lysate and compare it with a measurement of ATP in cell lysate treated with an inhibitor of ATPase or with an inhibitor of glycolysis. Thus the authors could clearly show that GOLPHE3 depletion is related to mitochondrial dysfunction. Internal control are sometimes lacking, for example: ROS production, a measurement performed on cell treated with an antioxydant should be added as well as for mitochondrial membrane potential, a decoupling agent should be used. Since the authors observe an important decrease of the mitochondrial membrane potential, an analysis of apoptosis or cell death could be appropriated.
Author Response
Reviewer #1:
1) This work is particularly interesting but some points should be clarified.
Response 1: We thank Reviewer #1 for this kind comment.
Reviewer #1:
2) The fluorescent background observed with GFP-GOLPHE3 in MDA-MB-231 cells is quite high. The authors should give some explanation or should present a picture with a lower exposition time.
Response 2: We thank Reviewer #1 for raising this important issue that we overlooked in our manuscript, but now we have addressed it by adding an explanation. GOLPH3 is a peripheral membrane protein of the Golgi apparatus with a large cytosolic pool (doi.org/10.1111/j.1600-0854.2000.11206.x; doi:10.1371/journal.pone.0154719), as we have stated in the section “Introduction” of our manuscript (lines 66-67). Importantly, GFP-GOLPH3 also behaves as a peripheral membrane protein of the Golgi apparatus with a large cytosolic pool (doi: 10.1091/mbc.e05-05-0447; doi:10.1371/journal.pone.0154719). Thus, it is expected that the fluorescent signal in the cytoplasm of cells expressing GFP-GOLPH3, besides that of the Golgi apparatus, corresponds to that of the cytosolic pool of GFP-GOLPH3 instead of being a high fluorescent background. Moreover, we have shown that different human breast cancer cell lines contain different proportions of endogenous GOLPH3 associated with the Golgi apparatus and the cytosol. Specifically, MDA-MB-231 cells contain a larger cytosolic pool of GOLPH3 compared to MCF7 cells (doi:10.1371/journal.pone.0154719). Likewise, MDA-MB-231 cells express GFP-GOLPH3 in the cytosol more than MCF7 cells (doi:10.1371/journal.pone.0154719). We have hypothesized that these differences in the distribution of GOLPH3 between the Golgi apparatus and the cytosol could be the consequence of different mechanisms of regulation in different types of cells, such as different post-translational modifications (doi:10.1371/journal.pone.0154719). Thus, the higher cytoplasmic fluorescent signal of MDA-MB-231 cells expressing GFP-GOLPH3, compared to HeLa cells, is very likely due to a similar phenomenon to that that we have found between MDA-MB-231 cells and MCF7 cells. In this regard, in our present report, we have consistently obtained higher cytoplasmic fluorescent signals in MDA-MB-231 cells expressing GFP-GOLPH3 compared to HeLa cells therefore presenting images with lower exposition time does not represent faithfully what we have consistently observed. Accordingly, we have modified the text of the corresponding “Results” section (lines 319-320 and lines 411-421) explaining the observed higher cytoplasmic fluorescent signal in MDA-MB-231 cells expressing GFP-GOLPH3.
Reviewer #1:
3) Fig 4 and fig 5 the legend should be completed. Indeed, the authors should explain their quantification: how many cells were counted from how many independent experiments?
Response 3: We thank this Reviewer for this comment. As requested, we have included in the legends of Figures 4 and 5 how many cells were used for the quantification and how many independent experiments were performed. In the case of Figure 4, we performed 3 independent experiments and quantified mitochondria profiles and mitochondria volume in 15-26 MDA-WT cells and 15-26 MDA-GOLPH3-KO cells. In the case of Figure 5, we performed 3 independent experiments and analyzed 6 MDA-WT cells and 6 MDA-GOLPH3-KO cells for each parameter.
Reviewer #1:
4) To clearly compare global ATP level with mitochondrial ATP it should be more relevant to measure the global ATP level in cell lysate and compare it with a measurement of ATP in cell lysate treated with an inhibitor of ATPase or with an inhibitor of glycolysis. Thus the authors could clearly show that GOLPH3 depletion is related to mitochondrial dysfunction.
Response 4: We thank this Reviewer for raising this issue, which we have addressed by performing additional analyses. As suggested, we quantified ATP levels in lysates obtained from cells treated with 10 or 20 μM oligomycin, an inhibitor of ATP synthase (doi:10.1111/febs.14756). We found that inhibiting ATP synthase significantly reduced ATP content in MDA-GOLPH3-KO cells compared to control cells, suggesting that GOLPH3 depletion could stimulate higher mitochondrial ATP production. Accordingly, we have modified the text describing the new data in the corresponding section of “Results” (lines 659-662), added the supplementary Figure 2 and its legend (panel A) to account for the new data, and modified the corresponding “Materials and Methods” section (lines 265-266).
Reviewer #1:
5) Internal controls are sometimes lacking, for example: ROS production, a measurement performed on a cell treated with an antioxidant should be added as well as for mitochondrial membrane potential, a decoupling agent should be used.
Response 5: As requested by this Reviewer, for ROS production, we performed an assay in which cells incubated with CM-H2DCFDA (DCF) were treated with MitoQ, a mitochondria-targeted antioxidant (doi:10.3389/fcell.2023.1048177), for 30 minutes, and quantified the fluorescence intensity before and after MitoQ administration using confocal microscopy on live cells. We observed a decrease in fluorescence intensity, which is indicative of a reduction in reactive oxygen species due to its antioxidant capacity. For the mitochondrial membrane potential assay, we utilized the uncoupling agent carbonyl cyanide 4-(trifluoromethoxy)phenylhydrazone (FCCP), widely used in mitochondrial membrane potential measurements (doi:10.3390/cells12071089). Cells were treated for 30 minutes with 40 μM FCCP, followed by incubation with TMRE to quantify mitochondrial membrane potential using a fluorimeter. Both MDA-WT and MDA-GOLPH3-KO cells treated with FCCP exhibited a significant decrease in fluorescence, suggesting a reduction in the mitochondrial membrane potential compared to control cells not treated with FCCP. We have modified the text describing the new data in the corresponding section of “Results” (lines 679-686 and lines 702-716), added panel B to the supplementary Figure 2 and its legend to account for the new data, and modified the corresponding “Materials and Methods” section (lines 283-286 and lines 293-296).
Reviewer #1:
6) Since the authors observe an important decrease of the mitochondrial membrane potential, an analysis of apoptosis or cell death could be appropriate.
Response 6: As this Reviewer suggested, we performed immunoblot analysis of the levels of proteins related to apoptosis, including caspase 9, caspase 3, cytochrome c and Bcl2. We did not observe significant changes in the levels of these proteins in MDA-GOLPH3-KO compared to MDA-WT, suggesting that apoptosis is not active in GOLPH3-depleted cells. We have modified the text describing the new data in the corresponding section of “Results” (lines 719-724), prepared a figure that we have included as supplementary material (supplementary Figure 3), and modified the corresponding “Materials and Methods” section to include the source of the antibodies used (lines 178-182).
Reviewer 2 Report
Comments and Suggestions for Authors
In this research, the authors evaluated the role of GOLPH3 in mitochondrial fission and is necessary to sustain bioenergetics function in MDA-MB-231 breast cancer cells. Generally, it’s meaningful and interesting research. In my opinion, the current version of this manuscript fits the scope of the Cells and could be accepted after minor revision.
My specific comments are in detail listed below:
1. Some references are of the wrong formats. Besides, some references are out of date (before 2010). Some recent ones may be better.
2. Recently, it was newly revealed that mitochondria dysfunction may affect the expression of immune-related proteins or pathways. Since GOLPH3 knockout or knockdown could induce mitochondrial fission in MDA-MB-231 breast cancer cells. Could this pathway also possess such function? The authors could discuss it in the discussion. Some references could be added to this part including 10.1016/j.jconrel.2022.11.004.
3. Some minor mistakes exist in this work. The authors may carefully check it.
4. In the introduction part or results part, the authors should discuss the recent development of mitochondrial dysfunction in inducing tumor cell death or affecting the immune status of tumors, which may raise the importance of this research rated to GOLPH3. Some references could be added to this part including 10.1002/advs.202207608.
5. If possible, some in vivo assay could be added, such as the anti-tumor capacity of GOLPH3 silencing.
6. How was the tumor cell growth affected by GOLPH3 silencing in combination with some other treatments? The authors should discuss it or prove it.
Author Response
Reviewer #2:
1) Generally, it’s meaningful and interesting research. In my opinion, the current version of this manuscript fits the scope of the Cells and could be accepted after minor revision.
Response 1: We appreciate this kind comment from Reviewer #2.
Reviewer #2:
2) Some references are of the wrong formats. Besides, some references are out of date (before 2010). Some recent ones may be better.
Response 2: As requested, we have corrected all references that were listed with the wrong format (former references 8, 57, and 58). However, we are not sure which references Reviewer #2 indicates are outdated. Of 18 references prior to 2010, 15 corresponded to original findings that have stood the test of time and that we have decided to preserve due to their relevance. Nevertheless, we have replaced some references that provided general background information with more recent references containing updated information. We have substituted the previously listed reference 4 (doi:10.1083/jcb.200105033) with the newly added reference 4 (doi: 10.1177/25152564231208250); previously listed reference 37 (doi: 10.1074/jbc.M708339200) with the newly added reference 37 (doi: 10.1016/j.plaphy.2007.12.012). In addition, we have removed the previously listed reference 36 (doi: 10.1096/fj.202100067R), and to account for all Reviewers’ requests, we added newly listed references 13, 46, 48, 52, 53, and 67-82
Reviewer #2:
3) Recently, it was newly revealed that mitochondria dysfunction may affect the expression of immune-related proteins or pathways. Since GOLPH3 knockout or knockdown could induce mitochondrial fission in MDA-MB-231 breast cancer cells. Could this pathway also possess such function? The authors could discuss it in the discussion. Some references could be added to this part including 10.1016/j.jconrel.2022.11.004.
Response 3: We appreciate this comment of Reviewer #2, and, as requested, we included in the section “Discussion” a paragraph (lines 912-933) that discusses the implication of GOLPH3 levels on mitochondrial dysfunction and its impact on immune-related proteins or pathways. However, we moderated the discussion of the implications of potential impacts only at the level of the PD-1/PD-L1 axis to minimize speculation. As requested, we have included reference 10.1016/j.jconrel.2022.11.004, along with additional references (67-82), which further substantiate our argument.
Reviewer #2:
4) Some minor mistakes exist in this work. The authors may carefully check it.
Response 4: We appreciate this comment from Reviewer #2, but we are not sure what kind of mistakes are being referred to. Nevertheless, we thoroughly revised the English language and modified the text on parts that needed to be clarified. These are minor modifications that are highlighted in our revised manuscript, as editorially requested.
Reviewer #2:
5) In the introduction part or results part, the authors should discuss the recent development of mitochondrial dysfunction in inducing tumor cell death or affecting the immune status of tumors, which may raise the importance of this research related to GOLPH3. Some references could be added to this part including 10.1002/advs.202207608.
Response 5: We thank Reviewer #2 for this suggestion, which we feel will strengthen our manuscript. As indicated in Response 3 to Reviewer #2, we included in the section “Discussion” a paragraph (lines 912-933) that discusses the implication of GOLPH3 levels on mitochondrial dysfunction, including a possible induction of tumor cell death or affecting the immune status of tumors. As requested, we added reference 10.1002/advs.202207608 and the references indicated in Response 3 to Reviewer #2.
Reviewer #2:
6) If possible, some in vivo assay could be added, such as the anti-tumor capacity of GOLPH3 silencing.
Response 6: We appreciate this suggestion of Reviewer #2. However, we think that performing in vivo experiments to test the anti-tumor capacity of GOLPH3 silencing is beyond the scope of our study, which was intended to test the possibility that GOLPH3 could be participating in the mitochondrial fission process proposed for Golgi-derived PI(4)P-containing vesicles. On the other hand, the cumulative experimental evidence strongly supports the tumorigenic capacity of GOLPH3 overexpression as well as the anti-tumor capacity of GOLPH3 silencing in several in vivo systems (doi: 10.1038/nature08109), including in breast cancer cells (doi: 10.1158/1078-0432.CCR-11-3156) and glioma (doi: 10.1007/s11060-018-2966-6; doi.org/10.1007/s00109-019-01843-4). Thus, to strengthen the notion of the anti-tumor capacity of GOLPH3, we added text and references to the “Discussion” section in the same paragraph as mentioned before (lines 912-933).
Reviewer #2:
7) How was the tumor cell growth affected by GOLPH3 silencing in combination with some other treatments? The authors should discuss it or prove it.
Response 7: Like our previous response (Response 6 to Reviewer #2), we think that performing tumor cell growth assays upon GOLPH3 silencing in combination with some other treatment is beyond the scope of our study. The potential capacity of combinatorial treatments using GOLPH3 silencing has already been tested and proved successful. Thus, to strengthen the notion of the potential anti-tumor capacity of GOLPH3 silencing in combination with some other treatments, we added text to the “Discussion” section (lines 912-933), together with new references.
Round 2
Reviewer 1 Report
Comments and Suggestions for Authors
Catalina M. Polanco and colleagues have improved their manuscript according to all comments, thus the article can be accepted in the present form. Thanks to all of them for their answer.
Author Response
Reviewer #1:
1) Catalina M. Polanco and colleagues have improved their manuscript according to all comments, thus the article can be accepted in the present form. Thanks to all of them for their answer.
Response 1: We thank Reviewer #1 for this kind comment.